# Should we allocate more COVID-19 vaccine doses to non-vaccinated individuals?

**Zied Ben Chaouch**[1,2], **Andrew W. Lo**[1,2,3,4]*, **Chi Heem Wong**[2,3,4]

**1** Department of Electrical Engineering and Computer Science, MIT, Cambridge, MA, United States of America, **2** Laboratory for Financial Engineering, MIT, Cambridge, MA, United States of America, **3** Sloan School of Management, MIT, Cambridge, MA, United States of America, **4** Computer Science and Artificial Intelligence Laboratory, MIT, Cambridge, MA, United States of America

* alo-admin@mit.edu

**Data Availability Statement:** We use publicly available data from the CDC and the John Hopkins University CCSE (https://github.com/CSSEGISandData/COVID-19).

## Abstract

Following the approval by the FDA of two COVID-19 vaccines, which are administered in two doses three to four weeks apart, we simulate the effects of various vaccine distribution policies on the cumulative number of infections and deaths in the United States in the presence of shocks to the supply of vaccines. Our forecasts suggest that allocating more than 50% of available doses to individuals who have not received their first dose can significantly increase the number of lives saved and significantly reduce the number of COVID-19 infections. We find that a 50% allocation saves on average 33% more lives, and prevents on average 32% more infections relative to a policy that guarantees a second dose within the recommended time frame to all individuals who have already received their first dose. In fact, in the presence of supply shocks, we find that the former policy would save on average 8, 793 lives and prevents on average 607, 100 infections while the latter policy would save on average 6, 609 lives and prevents on average 460, 743 infections.

## 1 Introduction

With more than 44.7 million infections in the U.S. and 219 million worldwide, and a death toll over 721,000 in the United States and 4.55 million worldwide, the COVID-19 pandemic has profoundly altered the research agenda of the scientific community as a whole, launching an unprecedented race against the clock to develop a cure or a vaccine for the disease.

To better contain the disease, and to design more efficient policies in combating it, the United States Centers for Disease Control and Prevention (CDC) has collected and combined an ensemble of models to forecast the spread of the epidemic [1]. These models range from traditional SIR and SEIR-type models, to agent-based models, mixture models, and machine-learning models. Some models, such as the DELPHI [2] model, explicitly account for the effects of government intervention, such as the implementation of social distancing policies. These models have quickly been applied practically: for example, at the clinical level, the DELPHI model helped to reallocate ventilators and alleviate shortages [3]; similarly, at the policy level, the DELPHI was used to propose a more efficient allocation of vaccines [4, 5]. Epidemiological models have also been used to optimize the design of vaccine clinical trials, and to

**Funding:** The authors received no specific funding for this work.

**Competing interests:** We have read the journal's policy and the authors of this manuscript have the following competing interests: ZBC and CHW report no conflicts. AWL reports personal investments in private biotechnology companies, biotechnology venture capital funds, and mutual funds. AWL is a co-founder and principal of QLS Advisors LLC, a healthcare investments advisor, and QLS Technologies LLC, a healthcare analytics and consulting company; an advisor to AlphaSimplex Group, Apricity Health, Aracari Bio, BrightEdge Impact Fund, Enable Medicine, FINRA, Lazard, Quantile Health, the Swiss Finance Institute, and Thalēs; a director of Annual Reviews, Atomwise, BridgeBio Pharma, and Roivant Sciences; and a member of the Board of Overseers at Beth Israel Deaconess Medical Center and the NIH's National Center for Advancing Translational Sciences Advisory Council and Cures Acceleration Network Review Board. During the most recent 6-year period, AWL has received speaking/consulting fees from: AlphaSimplex Group, Annual Reviews, Bernstein Fabozzi Jacobs Levy Award, BIS, BridgeBio Pharma, Cambridge Associates, CME, Financial Times, Harvard Kennedy School, IMF, JOIM, National Bank of Belgium, New Frontiers Advisors (for the 2020 Harry M. Markowitz Prize), Q Group, Research Affiliates, Roivant Sciences, and the Swiss Finance Institute. There are no patents, products in development or marketed products to declare. This does not alter our adherence to PLOS Global Public Health policies on sharing data and materials.

quantify the potential advantages of using adaptive Randomized Clinical Trials (RCTs) and Human Challenge Trials (HCTs) over traditional RCTs [6, 7]. Epidemiological models and simulations have helped researchers and policymakers answer pressing questions, such as how to prioritize the delivery of vaccine across demographics and medical conditions [8], and where should vaccination clinics be located to maximize the effectiveness of the vaccination campaign [4, 5].

With a rising number of infections and deaths, and the emergence of COVID-19 variants despite extended periods of lockdown, mass vaccination has become the critical pathway to alleviate the impact of the disease, as is apparent with the success of Israel's mass vaccination campaign [9]. However, producing and distributing the vaccines has become a new challenge for manufacturers. Despite promising results regarding the ability to store the Pfizer-BioNTech vaccine in standard freezers over periods of two weeks [10] rather than the initial storage constraint at −80˚C [11], vaccine shortages and appointment cancellations [12] have followed factory shutdowns [13], production mix-ups [14], delays in shipment [15], and power outages [16, 17]. Optimizing the allocation of vaccines has become crucial not only due to the limited supply of vaccines, but also due to the fact that Pfizer-BioNTech and Moderna vaccines need to be administered twice for each individual, over a recommended time interval of 3 or 4 weeks, respectively [18]. Although supply constraints are important in the United States, they are even more binding in other regions such as Canada [12, 15], Europe [13, 15, 19], Africa [20], Latin America [21], and India [22].

An important debate has also arisen regarding the advantages of delaying the second dose to provide more first doses to susceptible individuals [23–27]. While doses were held back under the Trump administration in order to guarantee a second dose to individuals who have received their first dose, the Biden administration has pledged to reverse this policy and release all available doses [28]. Other countries, such as the United Kingdom and Canada, have already adopted the policy of delaying the second dose up to three months [29, 30], and Singapore is currently considering delaying the second dose up to 12 weeks [31]. However, as Texas, Washington State, and Michigan experienced in mid-February 2021, releasing too many doses for first-time users could lead to delays for individuals eligible to receive their second dose (a "second-shot crunch") [32].

Researchers, medical doctors, and clinicians have provided arguments for and against delaying the second dose [33]. On the one hand, while allocating more first doses may initially slow down the spread of the infections, and ultimately reduce the number of deaths by allowing a bigger proportion of the population to have *some* immunity, it is possible that protection will degrade over time, and delaying the second dose may leave at-risk individuals inadequately protected. From a disease evolutionary perspective, partial immunization could also contribute to the selection of vaccine-resistant variants of SARS-CoV-2 [34]. This point is now even more relevant with the spread of the Delta variant, currently the predominant variant in the U.S., which is twice as contagious as the original strain of the virus, yet only modestly decreases the effectiveness of the two mRNA vaccines considered [35]. On the other hand, clinical trial results and data from the Israeli mass vaccination campaign on the efficacy of the first dose tend to support the policy of delaying the second dose up to three months, especially when the supply of vaccines is constrained [36–39].

In this work, we forecast the effect of various vaccine allocation strategies on the cumulative number of infections and deaths in the United States to quantify the impact of prioritizing first doses versus second doses. In particular, we extend the DELPHI model to account for vaccines, and use a simple model of shocks to the number of vaccines supplied to account for distributional constraints. Similar questions have recently been studied by other researchers. For example, [40, 41] recommend a second dose deferral strategy in order to vaccinate more

people faster even if the single-dose efficacy decays over time. Likewise, [42] using the agent-based epidemics model developed in [43], suggest a 9-week delay for the second dose, although the results are mixed for the Pfizer-BioNTech vaccine when the efficacy of the first dose decays over time. While our analysis focuses on the United States, our recommendations can be generalized to other countries and especially those where the supply of vaccines is heavily limited. Furthermore, the framework provided here can be reused in the event of a future pandemic to improve the allocation of vaccines and reduce the number of infections and deaths.

The remainder of the paper is structured as follows: we present the epidemiological model used to forecast the COVID-19 outbreak from October 1st, 2020 to August 1st, 2021 in Section 2, as well as the model used to account for supply shocks; our forecasts are presented in Section 3, and the policies under investigation are compared and discussed in Section 4; we conclude in Section 5. Finally, a more detailed description of our analysis is available in S1 Text.

## 2 Methodology

We begin by presenting the epidemiological model used to simulate the COVID-19 pandemic, the assumptions made in our forecasts, as well as the model used to simulate the supply of vaccine under random shocks.

### Epidemiological model

Many epidemiological models have been proposed to forecast the spread of COVID-19 [1]. In particular, [2] proposes a novel SEIR-based model, called the DELPHI model, that explicitly accounts for the effects of government intervention. As shown in Fig 1, the DELPHI model categorizes individuals into eight classes: Susceptible individuals who have not been infected ($S$); Exposed individuals who have been infected, and are currently within the incubation period ($E$); Infected and contagious individuals ($I$), who are then categorized into the Detected Hospitalized ($DH$), the Detected and home-Quarantined ($DQ$), and the Undetected and self-quarantined ($U$) classes; Recovered individuals ($R$); and individuals Deceased from the COVID-19 ($D$).

As we consider two hypothetical vaccines in this study (loosely modelled after the Moderna vaccine and the Pfizer-BioNTech vaccine), we augment the DELPHI model by including five vaccination categories for each vaccine brand $X$ used: individuals receiving their first dose who respond to the first dose ($V_{X,1}^{r}$ for immediate "response"), individuals receiving their first dose who do not respond to the first dose but will respond to the second dose ($V_{X,1}^{dr}$ for "delayed response"), and individuals receiving their first dose who will neither respond to the first dose nor to the second dose ($V_{X,1}^{nr}$ for "no response"); individuals who receive their second dose and respond to the vaccine ($V_{X,2}^{r}$); and individuals who receive their second dose and do not respond to the vaccine ($V_{X,2}^{nr}$).

We assume that the exposed individuals ($E$) are not yet contagious, and that recovered individuals ($R$) and vaccinated individuals from the $V_{X,2}^{r}$ group have permanent immunity to COVID-19. We further assume that the infection rate of individuals depends on a government response function (see Appendix A.1 in S1 Text) which models the effects of government intervention. The dynamics of the augmented DELPHI model are available in Appendix A.1 in S1 Text.

### Data and assumptions

The first step of the analysis consists in fitting the original DELPHI model to historical data using the dataset developed by [2]. After estimating the parameters of the original DELPHI

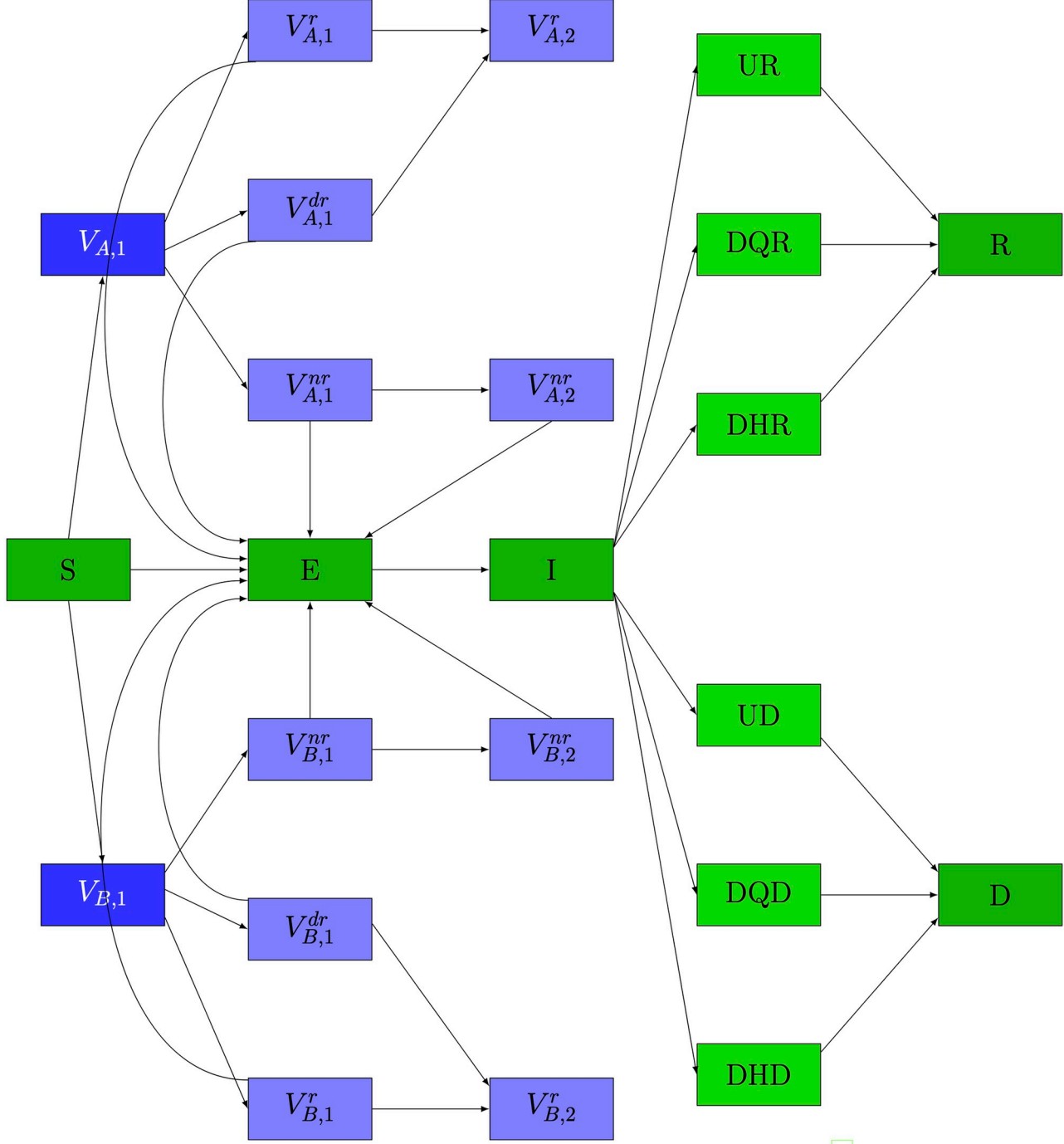

**Fig 1. Flowchart of the original DELPHI model (in green) [2] and the additional vaccination states (in blue) for two hypothetical vaccines.** For illustrative purposes, Vaccine A is loosely modelled after the Moderna vaccine, and Vaccine B after the Pfizer-BioNTech vaccine.

model for each state of the U.S., we recalibrate these parameters to allow us to simulate a discretized version of the DELPHI model using a time step of 1 day. We then ensure that the discretized model yields the same output as the original continuous-time model (see Appendix A.2 in S1 Text for more details). This step is crucial, as it considerably improves the speed of

**Table 1. Vaccination parameters used in the augmented DELPHI model for two hypothetical vaccines.** For illustrative purposes, Vaccine A is loosely modelled after the Moderna vaccine, and Vaccine B after the Pfizer-BioNTech vaccine.

| | Brand A | Brand B | Perturbations |
|---|---|---|---|
| 1st dose Efficacy[b] | 80.20% | 52% | ±20% |
| 2nd dose Efficacy[b] | 95.60% | 92% | +4%, −20% |
| Time Until 2nd dose[c] | 21 days | | 28, 35, 49, & 63 days |
| Time to Develop Permanent Immunity[a] | 14 days | | 21 days |
| Vaccination Start Date[d] | December 15th, 2020 | | |

[a] [44]

[b] [45]

[c] The FDA recommends 21 days for the Pfizer-BioNTech vaccine and 28 days for the Moderna vaccine. As shown in Appendix B.1 in S1 Text, this difference will have no impact on the analysis.

[d] [46]

the simulation and allows us to run Monte Carlo analyses. For consistency of the results, all simulations are based on the set of discretized parameters.

The parameters used in the augmented DELPHI model are presented in Table 1. We assume a uniform daily infection rate among individuals in each vaccination state. Individuals who respond to the first dose (the "immediate response" group $V_{X,1}^r$) remain completely susceptible to an infection in the first 14 days of their first vaccination, but become permanently immune to the disease 14 days following their first dose. Similarly, individuals in the "delayed response" group i.e., the $V_{X,1}^{dr}$ group, remain completely susceptible to an infection 35 days after receiving their first dose (i.e., 21 days to receive their second dose after their first dose, followed by 14 days to develop permanent immunity), but develop permanent immunity immediately afterwards. Individuals who neither respond to the first nor second dose i.e., the $V_{X,1}^{nr}$ group, remain permanently susceptible to an infection. Although the United States Food and Drug Administration (FDA) recommends a time interval between vaccine doses of 21 days for the Pfizer-BioNTech vaccine and 28 days for the Moderna vaccine, this difference has no impact on the analysis, as shown in Appendix B.1 in S1 Text. Finally, we assume that the immune response to a vaccine does not decay over time.

## Modeling the supply of vaccines

To explore the effect of shocks to the supply of vaccines on the vaccination policy adopted, we decompose the vaccine rollout into two phases: during the *ramp-up phase*, the number of new vaccine doses supplied increases at a linear rate, until it reaches a terminal value of 1.5 million new doses per day (President Biden's target [47]); This terminal value is reached on the 90th day, when we enter the *steady-state phase* in which the supply rate of new doses becomes constant. The assumed terminal value is on the conservative side, as the 7-day moving average of the number of doses administered daily (as reported to the CDC) increased from 1.5 million doses per day in February 2021 to 3 million in April 2021 [46, 48, 49] (a terminal rate of 3 million doses per day is explored in Appendix B.3 in S1 Text). The black curve in Fig 2 represents the daily number of new vaccine doses supplied by one vaccine company. As shown in the plot, the number of doses supplied by this company increases linearly, until it reaches a value of 0.75 million doses (one half of 1.5 million, as we consider two vaccines in this study).

To model supply shocks, we assume that shock occurrences follow a Poisson process with a rate of 1 shock per 30 days. Using a Poisson process is appropriate here as we assume that

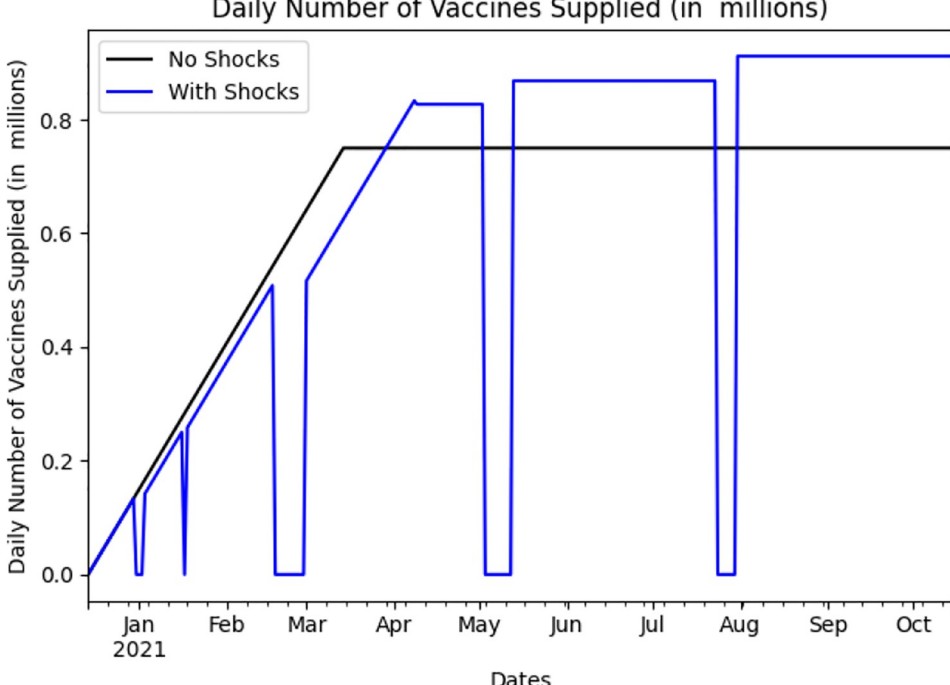

**Fig 2. Example of the daily number of vaccines supplied by one company with and without supply shocks between December 15th, 2020 and August 1st, 2021.**

shocks that occur over disjoint time intervals are independent, and that the process is memory-less. Once a shock occurs, the supply of this particular vaccine drops to zero over a length of time drawn from a uniform distribution between 0 and 14 days. The supply then picks up at the previous positive level and continues to increase linearly. Furthermore, we assume that shocks lasting 7 days or more have a 50% probability of boosting the terminal supply rate by 5%. The blue curve in Fig 2 provides an example of a supply curve with such shocks. Supply shocks can simply represent delays in production or delivery of vaccines (which would tend to last a few days), but they can also model a factory shutdown aimed at improving the production of vaccines (which would tend to last longer and may increase the terminal supply rate).

Finally, each state receives a fraction of the number of available doses in proportion to its population size. We then run Monte Carlo simulations to investigate the robustness of the vaccination policy to supply shocks. The number of cumulative deaths and cumulative infections are aggregated at the country level, and are used to compare vaccination policies.

## 3 Results

In this section, we explore the performance of various vaccination policies, and evaluate them based on the number of cumulative deaths and cumulative infections aggregated at the country level. This helps us understand whether we should store vaccine doses in order to guarantee a second dose to individuals who received a first dose, or if it is more efficient to allocate as many first doses as possible.

Storing doses ensures that individuals who received a first dose will be able to obtain their second dose according to the recommended vaccination schedule (here, 21 days) even if supply shocks occur. However, this strategy reduces the number of individuals that can be

vaccinated each day, and may lead to a higher cumulative number of deaths and infections. We further assume that 1% of unused doses are lost each day in order to model spoilage or wastage due to unforeseen circumstances.

## Vaccination policies

The policies we investigate are described below.

**Baseline policy.** As a baseline policy, we consider the case of not vaccinating the population. This case is expected to present the highest number of cumulative infections and deaths.

**Policy of interest.** The vaccination policy of interest consists in allocating a fixed fraction of available doses to first-time users, and allocating the remaining doses to individuals who have already received their first dose and are eligible to receive their second dose. Furthermore, unused doses are reallocated to individuals eligible to get a vaccine. For example, under a policy of interest allocating 75% of doses to first-time users, 75% of the doses available today would be administered to individuals who have not received their first dose and 25% of doses will be administered to individuals who have received their first dose at least 21 days ago; if doses are unused because we have more second doses available today than eligible individuals for a second dose, we reallocate these unused doses to first-time users; if doses are unused because we have more first doses available today than individuals eligible for their first dose, these unused doses are reallocated to individuals eligible for a second dose today. In comparison, we also consider a scenario under which we do not allow for doses reallocation.

The policy of interest is then compared to the following alternatives.

**Alternative policy I: Strong priority scenario.** Doses are allocated by prioritizing all individuals who have received a first dose and will eventually need to receive the second dose in the future. This means that all individuals who receive their first dose are guaranteed to receive their second dose within the recommended time frame. Under the strong priority scenario, a second dose will immediately be placed in storage each time an individual receives their first dose, and this dose will be administered to this individual 21 days later.

**Alternative policy II: Weak priority scenario.** In contrast to the strong priority scenario, this policy consists of allocating doses in priority to individuals scheduled to receive their second dose on that specific day. In the weak priority scenario, doses available today are first administered to individuals eligible to receive their second dose; after clearing the second dose queue, the remaining doses are allocated to first-time users.

In all the vaccination policies described above, second doses are always allocated in a First-In-First-Out (FIFO) fashion: this gives higher priority to individuals eligible for a second dose who have not been able to receive their second dose yet over individuals who only became eligible for a second dose today.

A final point: it can be useful to view the weak priority scenario as a special case of the policy of interest in which we reallocate unused doses. In fact, under a policy of interest that allocates 0% of doses towards first-time users, individuals eligible to receive their second dose today will be given priority; then unused doses would be reallocated towards first-time users.

## Policy evaluation

To compare the four policies described in Section 3, we simulate the evolution of the epidemics in the absence and in the presence of random supply shocks. In particular, we run 1,000 simulations to obtain a distribution for the cumulative number of infections and the cumulative number of deaths between October 1st, 2020 and August 1st, 2021 under random supply shocks. After comparing the outputs obtained with various number of Monte Carlo simulations, we selected the number of Monte Carlo simulations to be large enough to reflect the

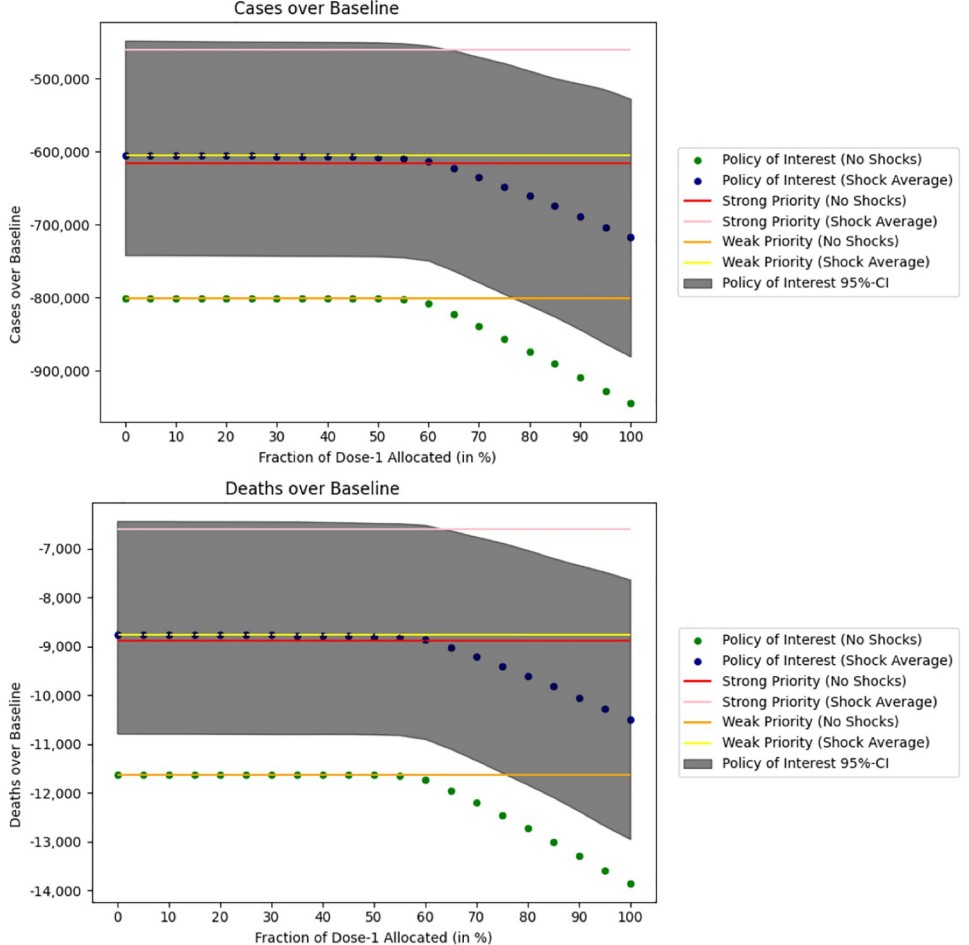

**Fig 3. Simulation of the DELPHI model under supply shocks.** We calculate the **cumulative number of deaths and infections** between October 1st, 2020 and August 1st, 2021 relative to a no vaccination baseline when a constant fraction of available doses are allocated to first-time users. Results under supply shocks are averaged over 1,000 Monte Carlo simulations. We use the February 7th, 2021 DELPHI model parameters.

uncertainty in the output while being parsimonious enough to retain a practical simulation runtime. The DELPHI parameters used in all our forecasts were estimated on February 7th, 2021.

We plot in Fig 3 the cumulative number of infections and the cumulative number of deaths under the policy of interest in the absence of supply shocks (green dots) and in the presence of supply shocks (blue dots) as we increase the allocation of doses towards first-time users. The numbers on the y-axis are negative as we display the number of infections and the number of deaths under the policy of interest relative to the baseline policy. In particular, we observe that allocating 50% of available doses to first-time users saves 11, 632 lives on average if there are no supply shocks, and saves an average of 8, 793 lives (95%CI: [6, 477;10, 803]) under random supply shocks relative to a no-vaccination policy. Allocating 50% of doses to first-time uses also prevents 801, 451 infections under no supply shocks and 607, 100 infections (95%CI: [450, 024;743, 198]) under supply shocks.

As we increase the fraction of doses allocated to first-time users, our forecasts predict a decrease in the cumulative number of infections and deaths. In fact, if we allocate all available

**Table 2. Simulation of the DELPHI model under supply shocks as we vary the fraction of doses allocated to first-time users.** We calculate the **number of infections** relative to a no-vaccination baseline when a constant fraction of available doses are allocated to first-time users. Results under supply shocks are averaged over 1,000 Monte Carlo simulations. We use the 2021/02/07 DELPHI model parameters.

| Dose 1 | No Shocks | Shock Average | Shock SD | Shock s.e. | 5th-perc. | 95th-perc. |
|---|---|---|---|---|---|---|
| 0 | -801,387 | -604,926 | 90,931 | 2,875 | -741,807 | -447,624 |
| 5 | -801,387 | -605,019 | 90,908 | 2,875 | -741,879 | -447,886 |
| 10 | -801,387 | -605,122 | 90,883 | 2,874 | -741,998 | -448,173 |
| 15 | -801,387 | -605,238 | 90,854 | 2,873 | -742,126 | -448,502 |
| 20 | -801,387 | -605,370 | 90,822 | 2,872 | -742,261 | -448,861 |
| 25 | -801,387 | -605,522 | 90,788 | 2,871 | -742,416 | -448,995 |
| 30 | -801,387 | -605,703 | 90,750 | 2,870 | -742,592 | -449,121 |
| 35 | -801,387 | -605,924 | 90,706 | 2,868 | -742,722 | -449,259 |
| 40 | -801,387 | -606,205 | 90,648 | 2,867 | -742,795 | -449,441 |
| 45 | -801,387 | -606,576 | 90,575 | 2,864 | -742,879 | -449,673 |
| 50 | -801,451 | -607,100 | 90,494 | 2,862 | -743,198 | -450,024 |
| 55 | -802,656 | -608,261 | 90,495 | 2,862 | -744,474 | -451,435 |
| 60 | -808,134 | -612,442 | 90,999 | 2,878 | -749,287 | -454,678 |
| 65 | -822,703 | -622,631 | 93,180 | 2,947 | -762,924 | -461,109 |
| 70 | -839,272 | -634,693 | 95,480 | 3,019 | -779,156 | -470,017 |
| 75 | -856,478 | -647,566 | 97,751 | 3,091 | -795,018 | -478,389 |
| 80 | -873,507 | -660,804 | 99,898 | 3,159 | -810,243 | -488,860 |
| 85 | -890,732 | -674,248 | 102,115 | 3,229 | -825,871 | -499,160 |
| 90 | -908,812 | -688,298 | 104,511 | 3,305 | -843,509 | -506,485 |
| 95 | -927,952 | -703,192 | 107,090 | 3,386 | -862,812 | -514,555 |
| 100 | -944,717 | -716,579 | 109,442 | 3,461 | -880,397 | -527,296 |

doses to first-time users, the policy of interest would save 13, 859 lives in the absence of supply shocks, and save on average 10, 497 lives (95%CI: [7, 634;12, 943]) under supply shocks. This policy would also prevent 944, 717 infections in the absence of supply shocks, and prevent on average 716, 579 infections (95%CI: [527, 296;880, 397]) under supply shocks. The estimates are reported in Tables 2 and 3.

In contrast, the alternative policies considered do not allocate a fixed fraction of available doses to first-time users. Instead, under the strong priority policy (Table 4), a second dose is kept in storage as soon as an individual receives his or her first dose. This strategy is able to save only 8, 876 lives under no supply shocks, and on average 6, 609 lives (95%CI: [4, 790;8, 213]) under supply shocks, while it prevents 616, 315 infections in the absence of supply shocks and on average 460, 743 infections (95%CI: [337, 131;569, 924]) under supply shocks.

The weak priority policy (Table 5) relaxes this restriction and distributes the available doses each day in priority to individuals who already received their first dose and are eligible to receive their second dose on that day. Under this strategy, our forecasts predict that 11, 631 lives would be saved under no supply shocks, and on average 8, 759 lives (95%CI: [6, 434;10, 786]) would be saved under supply shocks. This strategy would also prevent 801, 387 infections in the absence of supply shocks and on average 604, 926 infections (95%CI: [447, 624;741, 807]) under supply shocks.

It is important to highlight here that under the policy of interest, the cumulative number of infections and deaths remains constant as we vary the fraction of doses allocated to first-time users from 0% to about 50%. This effect is due to the reallocation of unused doses modelled by our simulation. More concretely, if we allocate no doses to first-time users (i.e., we only give

**Table 3. Simulation of the DELPHI model under supply shocks as we vary the fraction of doses allocated to first-time users.** We calculate the **number of deaths** relative to a no-vaccination baseline when a constant fraction of available doses are allocated to first-time users. Results under supply shocks are averaged over 1,000 Monte Carlo simulations. We use the 2021/02/07 DELPHI model parameters.

| Dose 1 | No Shocks | Shock Average | Shock SD | Shock s.e. | 5pct | 95pct |
|---|---|---|---|---|---|---|
| 0 | -11,631 | -8,759 | 1,349 | 43 | -10,786 | -6,434 |
| 5 | -11,631 | -8,760 | 1,349 | 43 | -10,789 | -6,435 |
| 10 | -11,631 | -8,762 | 1,349 | 43 | -10,789 | -6,435 |
| 15 | -11,631 | -8,764 | 1,348 | 43 | -10,790 | -6,437 |
| 20 | -11,631 | -8,766 | 1,348 | 43 | -10,793 | -6,438 |
| 25 | -11,631 | -8,768 | 1,347 | 43 | -10,795 | -6,438 |
| 30 | -11,631 | -8,771 | 1,347 | 43 | -10,798 | -6,441 |
| 35 | -11,631 | -8,774 | 1,346 | 43 | -10,799 | -6,441 |
| 40 | -11,631 | -8,779 | 1,345 | 43 | -10,797 | -6,453 |
| 45 | -11,631 | -8,785 | 1,344 | 42 | -10,798 | -6,463 |
| 50 | -11,632 | -8,793 | 1,342 | 42 | -10,803 | -6,477 |
| 55 | -11,648 | -8,810 | 1,342 | 42 | -10,817 | -6,484 |
| 60 | -11,730 | -8,870 | 1,350 | 43 | -10,895 | -6,517 |
| 65 | -11,946 | -9,024 | 1,383 | 44 | -11,098 | -6,628 |
| 70 | -12,198 | -9,205 | 1,418 | 45 | -11,334 | -6,756 |
| 75 | -12,454 | -9,400 | 1,453 | 46 | -11,585 | -6,878 |
| 80 | -12,723 | -9,605 | 1,488 | 47 | -11,831 | -7,027 |
| 85 | -12,996 | -9,819 | 1,524 | 48 | -12,087 | -7,192 |
| 90 | -13,286 | -10,045 | 1,562 | 49 | -12,370 | -7,337 |
| 95 | -13,593 | -10,285 | 1,603 | 51 | -12,675 | -7,473 |
| 100 | -13,859 | -10,497 | 1,640 | 52 | -12,943 | -7,634 |

doses to individuals who have already received their first dose) and *do not* reallocate unused doses, then nobody would ever receive their first dose and hence nobody will ever be eligible to receive a second dose. Reallocating unused doses overcomes this issue. Furthermore, reallocating unused doses under a 0% first dose allocation policy exactly matches the outcome of the weak priority scenario (the orange and yellow lines in Fig 3), in which we always give priority to individuals who already received their first dose and are now eligible to receive their second dose.

**Table 4. Simulation of the DELPHI model under supply shocks as we vary the fraction of doses allocated to first-time users.** We calculate the number of infections and deaths relative to a no-vaccination baseline under a **strong priority scenario**. Results under supply shocks are averaged over 1,000 Monte Carlo simulations. We use the 2021/02/07 DELPHI model parameters.

| | Cumulative Deaths | Cumulative Infections | Deaths over Baseline | Cases over Baseline |
|---|---|---|---|---|
| Setting | | | | |
| No Vaccination | 610,626 | 30,877,115 | 0 | 0 |
| No Shocks | 601,750 | 30,260,800 | -8,876 | -616,315 |
| Shock Average | 604,017 | 30,416,372 | -6,609 | -460,743 |
| Shock SD | 1,054 | 71,403 | 1,054 | 71,403 |
| Shock s.e. | 33 | 2,258 | 33 | 2,258 |
| t-Statistic | – | – | -198 | -204 |
| 5pct | 602,413 | 30,307,191 | -8,213 | -569,924 |
| 95pct | 605,836 | 30,539,984 | -4,790 | -337,131 |

**Table 5. Simulation of the DELPHI model under supply shocks as we vary the fraction of doses allocated to first-time users.** We calculate the number of infections and deaths relative to a no-vaccination baseline under a **weak priority scenario**. Results under supply shocks are averaged over 1,000 Monte Carlo simulations. We use the 2021/02/07 DELPHI model parameters.

| | Cumulative Deaths | Cumulative Infections | Deaths over Baseline | Cases over Baseline |
|---|---|---|---|---|
| Setting | | | | |
| No Vaccination | 610,626 | 30,877,115 | 0 | 0 |
| No Shocks | 598,995 | 30,075,728 | -11,631 | -801,387 |
| Shock Average | 601,867 | 30,272,189 | -8,759 | -604,926 |
| Shock SD | 1,349 | 90,931 | 1,349 | 90,931 |
| Shock s.e. | 43 | 2,875 | 43 | 2,875 |
| t-Statistic | – | – | -205 | -210 |
| 5pct | 599,840 | 30,135,308 | -10,786 | -741,807 |
| 95pct | 604,192 | 30,429,491 | -6,434 | -447,624 |

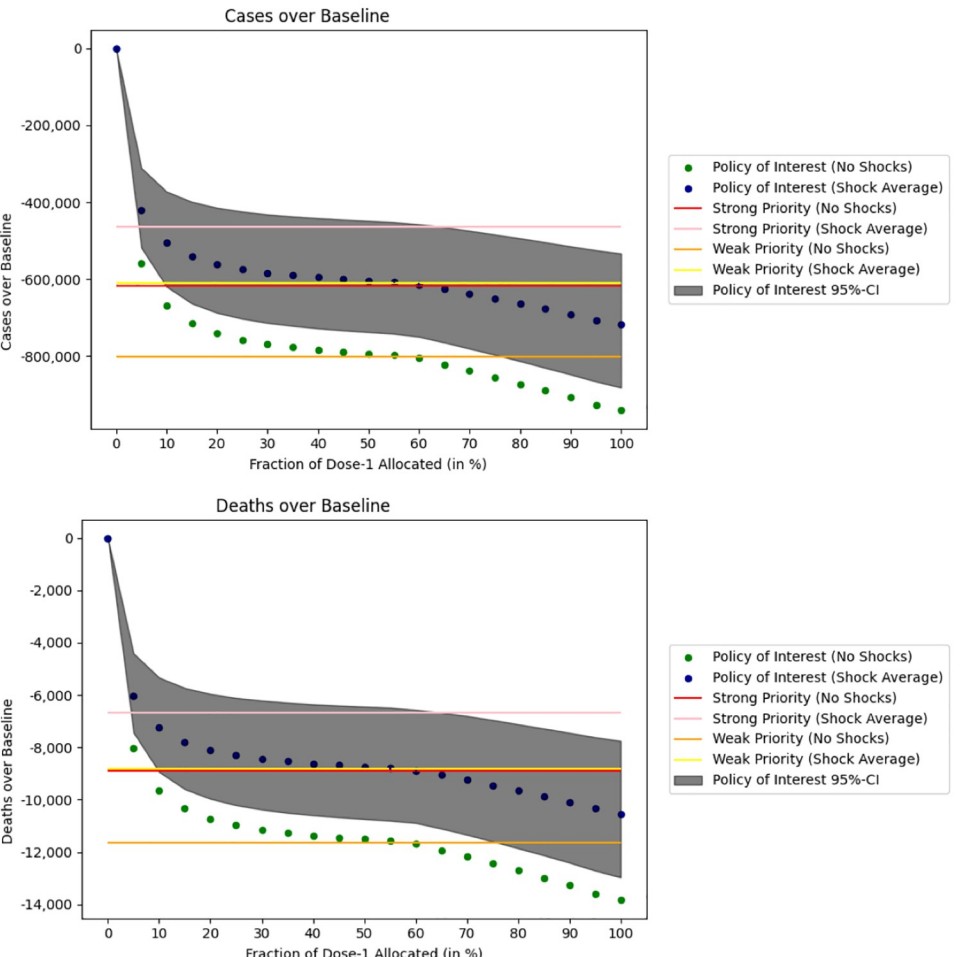

**Fig 4. Simulation of the DELPHI model under supply shocks when we do not reallocate excess doses to individuals eligible to get a vaccine.** We calculate the cumulative number of deaths and infections between October 1st, 2020 and August 1st, 2021 relative to a no vaccination baseline when a constant fraction of available doses are allocated to first-time users. Results under supply shocks are averaged over 1,000 Monte Carlo simulations. We use the February 7th, 2021 DELPHI model parameters.

When unused doses are not reallocated, we obtain the forecasts displayed in Fig 4. Allocating no doses to first-time users is identical to the no-vaccination policy, while increasing the dose allocation to first-time users beyond 30% produces similar results to Fig 3.

## 4 Discussion

### Policy comparison

As expected, vaccinating the population significantly reduces the number of infections and deaths under all the policies considered. However, the forecasts presented in Section 3 allow us to immediately rule out the efficiency of the first alternative policy (i.e., the strong priority scenario) relative to the other vaccination policies presented. In fact, both the policy of interest and the weak priority scenario are significantly better than the strong priority scenario in the presence and absence of random supply shocks. This is also expected, as more individuals are able to receive their first dose under the policy of interest and the weak priority scenario and start to develop an immune response early. The magnitude of the improvement is even more striking: under supply shocks, the policy of interest allocating 50% of available doses to first-time users is expected to save on average 33% more lives and prevents on average 32% more infections than the strong priority scenario. Nevertheless, the strong priority scenario is still important to analyze, as individuals getting a vaccine in the U.S. usually obtain an appointment for their second dose as soon as they receive their first dose, and patients requiring a second dose are given priority [50].

In the absence of supply shocks, the weak priority scenario is dominated by the policy of interest when more than 60% of available doses are allocated to first-time users. In particular, if we compare the weak priority scenario to a policy of interest allocating 85% of available doses to first-time users, our forecasts predict an increase in the number of lives saved and of the number of infected averted of 11.7% and 11.1% respectively. This result also holds in the presence of supply shocks: we forecast an increase of 12.1% in the number of lives saved and an increase of 11.5% in the number of infections averted. These differences are statistically significant as a Welch t-test yields t-statistics of 16.4 and 16, respectively, for the number of lives saved and infections averted. This is also expected, as the weak priority scenario can be viewed as a special case of the policy of interest, where less than 50% of available doses are allocated to first-time users, with unused doses being reallocated. As a consequence, the number of lives saved and infections averted will always be higher under the policy of interest.

In summary, our forecasts suggest that allocating more than 50% of available doses towards first-time users, even at the cost of delaying the distribution of second doses, would be a better policy than guaranteeing a second dose within the recommended time frame to every individual receiving their first dose. The simulations also show that these results are robust to supply shocks. Although our analysis focuses on the United States, the forecasts and their interpretation can be generalized to any country. Prioritizing first doses would be even more relevant to countries where the vaccine supply is severely limited (as shown in Appendix B.3 in S1 Text).

### Limitations and sensitivity analysis

Our forecasts are all based on an augmented version of the DELPHI epidemic model [2] that accounts for vaccinations. We should note that the model fails to account for demographics to assign different contact rates, hospitalization rates, and mortality rates across different age groups. Furthermore, some simplifying assumptions are used: for example, recovered individuals are assumed to have permanent immunity. However, among the top 10 models used by the CDC, DELPHI often displays the best performance with a low mean absolute percentage error (see https://www.covidanalytics.io/projections).

A critical limitation of our model is that we assume no decay in the efficacy of the vaccine over time if an individual has received their first dose, but are still waiting for their second dose. At this point, this decay in efficacy remains an open question [51]. Although our simulations begin on October 1st, 2020 and end on August 1st, 2021, vaccinations start on December 15th, 2020. If we consider a policy of interest that allocates 100% of available doses to first-time users, it would mean that individuals receiving their first dose at the end of December 2020 would not receive their second dose by August 1st, 2021. If the efficacy of the first dose decays over time, our forecast would be overly optimistic. However, knowing this decay rate would help determine the optimal fraction of doses that need to be allocated to first-time users under the policy of interest to balance the advantages of delaying the second dose against the efficiency loss due to the delay.

Finally, we find that our results remain significant as we perturb some key assumed parameters. We show in Appendix B in S1 Text the forecasts obtained as we increase the time interval between the first and second dose (from 4 weeks to 9 weeks), as we increase or decrease the efficacy of each vaccine dose, as we increase the time needed to develop permanent immunity, and as we increase the supply of vaccines. In particular, we observe that the curves obtained in Appendix B in S1 Text tend to shift upwards as we increase the time interval between the doses, increase the time needed to develop permanent immunity, decrease the supply of vaccines, or decrease the efficacy of each vaccine dose, implying an overall reduction in the number of lives saved and infections averted. In addition, the curves become flatter, implying a lower sensitivity to the chosen fraction of available doses allocated to first-time users, especially as we increase the time needed to develop permanent immunity, decrease the supply of vaccines, or decrease the efficacy of the first dose.

## 5 Conclusion

We have developed a systematic framework to compare the efficiency of various vaccination policies. In particular, we extend the DELPHI model [2] to account for vaccination states, and explore the impact of prioritizing vaccines to first-time users instead of guaranteeing a second dose within the recommended time frame to individuals who have already received their first dose.

Our forecasts suggest that allocating more than 50% of available doses to first-time users significantly increases the number of lives saved and significantly reduces the number of COVID-19 infections. It is important to highlight here that our forecasts are *not* recommending individuals to skip the second dose, a trend that has already raised some concerns as the efficacy of a single dose of mRNA vaccine over a long period of time remains unclear [52–54]. Instead, we suggest delaying the second dose to allow more individuals to receive the first dose in order to reduce the spread of the disease faster.

## Supporting information

**S1 Text. Appendix A: Dynamics of the augmented DELPHI model**. We describe here the additions made to the DELPHI model to include vaccination states as well as our discretization technique used to enhance the performance of the simulation. Appendix B: Sensitivity Analysis. We explore the sensitivity of our results to key parameters of the model and provide additional simulation results.
(PDF)

## Acknowledgments

We thank Jayna Cummings for editorial support. The views and opinions expressed in this article are those of the authors only, and do not necessarily represent the views and opinions of any institution or agency, any of their affiliates or employees, or any of the individuals acknowledged above. Research support from the MIT Laboratory for Financial Engineering is gratefully acknowledged.

## Author Contributions

**Conceptualization:** Zied Ben Chaouch, Andrew W. Lo, Chi Heem Wong.

**Data curation:** Zied Ben Chaouch, Chi Heem Wong.

**Formal analysis:** Zied Ben Chaouch, Chi Heem Wong.

**Funding acquisition:** Andrew W. Lo.

**Investigation:** Zied Ben Chaouch, Andrew W. Lo, Chi Heem Wong.

**Methodology:** Zied Ben Chaouch, Andrew W. Lo, Chi Heem Wong.

**Project administration:** Andrew W. Lo.

**Resources:** Andrew W. Lo.

**Software:** Zied Ben Chaouch, Chi Heem Wong.

**Supervision:** Andrew W. Lo.

**Validation:** Zied Ben Chaouch, Andrew W. Lo, Chi Heem Wong.

**Visualization:** Zied Ben Chaouch, Chi Heem Wong.

**Writing – original draft:** Zied Ben Chaouch, Andrew W. Lo, Chi Heem Wong.

**Writing – review & editing:** Zied Ben Chaouch, Andrew W. Lo, Chi Heem Wong.

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
