## [Decision Letter · Decision Letter 0]

25 Mar 2022

PGPH-D-21-00899

Should we allocate more COVID-19 vaccine doses to non-vaccinated individuals?

Dear Dr. Lo,

Thank you for submitting your manuscript to PLOS Global Public Health. After careful consideration, we feel that it has merit but does not fully meet PLOS Global Public Health’s publication criteria as it currently stands. Therefore, we invite you to submit a revised version of the manuscript that addresses the points raised during the review process.

We look forward to receiving your revised manuscript.

Kind regards,

Habib Hasan Farooqui, MD

Academic Editor

Journal Requirements:

1. We ask that a manuscript source file is provided at Revision. Please upload your manuscript file as a .doc, .docx, .rtf or .tex. If you are providing a .tex file, please upload it under the item type ‘LaTeX Source File’ and leave your .pdf version as the item type ‘Manuscript’.

2. Please provide separate figure files in .tif or .eps format only, and remove any figures embedded in your manuscript file.  If you are using LaTeX, you do not need to remove embedded figures.

3. Please ensure that the Title in your manuscript file and the Title provided in your online submission form are the same.

4. Please update the completed 'Competing Interests' statement, including any COIs declared by your co-authors. If you have no competing interests to declare, please state "The authors have declared that no competing interests exist". Otherwise please declare all competing interests beginning with the statement "I have read the journal's policy and the authors of this manuscript have the following competing interests:"

5. We have noticed that you have uploaded supporting information but you have not included a list of legends.  Please add a full list of legends for all supporting information files (including figures, table and data files) after the references list. 

6. Please amend your detailed Financial Disclosure statement. This is published with the article, therefore should be completed in full sentences and contain the exact wording you wish to be published.

i) Please include all sources of funding (financial or material support) for your study. List the grants (with grant number) or organizations (with url) that supported your study, including funding received from your institution. 

ii). State the initials, alongside each funding source, of each author to receive each grant.

iii). State what role the funders took in the study. If the funders had no role in your study, please state: “The funders had no role in study design, data collection and analysis, decision to publish, or preparation of the manuscript.”

iv). If any authors received a salary from any of your funders, please state which authors and which funders.

7. Please include details in the Funding Information in the system just like in the Financial Disclosure Statement.

Additional Editor Comments (if provided):

Reviewer 1

Authors have discussed various vaccination distributions policies to have a better effect in COVID control. They analyzed that allocating more than 50% of available doses to individuals who have not received their first dose can significantly increase the number of lives saved and significantly reduce the number of COVID-19 infections relative to a policy that guarantees a second dose within the recommended time frame to all individuals who have already received their first dose.

Reviewer 2

This study on vaccine supply and management is very important study when there has been frequent emergence of variants of Sars-Cov-2 virus due to large proportion of global population still remaining unvaccinated against Covid19. This study generates the evidence for efficiency of vaccination policy. Allocating 50% vaccines for first time users and vaccinating them may significantly reduce the deaths as well spread of Covid19. These results are in concurrence to ongoing discussion and research on partially vaccinating the majority of global population will reduce the emergence of variants.

Authors may give reasons for their assumptions such as why DELPHI model is discretized rather than working with ODE (continuous) model (Page 5), choice of Poisson distribution to model vaccine supply shocks (Page 7), carrying out only 1000 iterations of Monte Carlo simulation (Page 7), shocks lasting 7days or more is assumed to have 50%probability of boosting the terminal supply rate by 5%. Giving reason for all these assumptions will be strengthen the study from the point of view of research and will be helpful for the early researchers.

Reviewers' comments:

Reviewer's Responses to Questions

**Comments to the Author**

1. Does this manuscript meet PLOS Global Public Health’s publication criteria? Is the manuscript technically sound, and do the data support the conclusions? The manuscript must describe methodologically and ethically rigorous research with conclusions that are appropriately drawn based on the data presented.

Reviewer #1: Yes

Reviewer #2: Yes

2. Has the statistical analysis been performed appropriately and rigorously?

Reviewer #1: N/A

Reviewer #2: Yes

3. Have the authors made all data underlying the findings in their manuscript fully available (please refer to the Data Availability Statement at the start of the manuscript PDF file)?

Reviewer #1: Yes

Reviewer #2: Yes

4. Is the manuscript presented in an intelligible fashion and written in standard English?

Reviewer #1: Yes

Reviewer #2: Yes

5. Review Comments to the Author

Reviewer #1: Authors have discussed various vaccination distributions policies to have a better effect in COVID control. They analyzed that allocating more than 50% of available doses to individuals who have not received their first dose can significantly increase the number of lives saved and significantly reduce the number of COVID-19 infections relative to a policy that guarantees a second dose within the recommended time frame to all individuals who have already received their first dose.

This is an useful work, and the questions has been analyzed properly. I recommend for acceptance in its current form.

Reviewer #2: This study on vaccine supply and management is very important study when there has been frequent emergence of variants of Sars-Cov-2 virus due to large proportion of global population still remaining unvaccinated against Covid19. This study generates the evidence for efficiency of vaccination policy. Allocating 50% vaccines for first time users and vaccinating them may significantly reduce the deaths as well spread of Covid19. These results are in concurrence to ongoing discussion and research on partially vaccinating the majority of global population will reduce the emergence of variants.

Authors may give reasons for their assumptions such as why DELPHI model is discretized rather than working with ODE (continuous) model (Page 5), choice of Poisson distribution to model vaccine supply shocks (Page 7), carrying out only 1000 iterations of Monte Carlo simulation (Page 7), shocks lasting 7days or more is assumed to have 50%probability of boosting the terminal supply rate by 5%. Giving reason for all these assumptions will be strengthen the study from the point of view of research and will be helpful for the early researchers.

6. PLOS authors have the option to publish the peer review history of their article (what does this mean?). If published, this will include your full peer review and any attached files.

**Do you want your identity to be public for this peer review?** For information about this choice, including consent withdrawal, please see our Privacy Policy.

Reviewer #1: **Yes: **Samit Bhattacharyya

Reviewer #2: No

---

## [Editor Report · Decision Letter 1]

27 Apr 2022

Should we allocate more COVID-19 vaccine doses to non-vaccinated individuals?

PGPH-D-21-00899R1

Dear Dr Lo,

We are pleased to inform you that your manuscript 'Should we allocate more COVID-19 vaccine doses to non-vaccinated individuals?' has been provisionally accepted for publication in PLOS Global Public Health.

Best regards,

Habib Hasan Farooqui, MD

Academic Editor